# Herbal medicine (Hyeolbuchukeo-tang or Xuefu Zhuyu decoction) for treating primary dysmenorrhoea: protocol for a systematic review of randomised controlled trials

Junyoung Jo,[1] Jungtae Leem,[2,3] Jin Moo Lee,[4] Kyoung Sun Park[4]

► Prepublication history and additional material are available. To view these files please visit the journal online (http://dx.doi.org/ 10.1136/bmjopen-2016-015056)

JJ and JL contributed equally. JJ and JL are co-first authors.

[1]Department of Korean Medicine Obstetrics and Gynecology, Conmaul Hospital of Korean Medicine, Seoul, South Korea
[2]Korean Medicine Clinical Trial Center, Kyung Hee University Korean Medicine Hospital, Seoul, South Korea
[3]Department of Clinical Korean Medicine, Graduate School, Kyung Hee University, Seoul, South Korea
[4]Department of Korean Medicine Obstetrics and Gynecology, College of Korean Medicine, Kyung Hee University, Seoul, South Korea

**Correspondence to**
Dr Kyoung Sun Park;
lovepks0116@gmail.com

## ABSTRACT

**Introduction** Primary dysmenorrhoea is menstrual pain without pelvic pathology and is the most common gynaecological condition in women. Xuefu Zhuyudecoction (XZD) or Hyeolbuchukeo-tang, a traditional herbal formula, has been used as a treatment for primary dysmenorrhoea. The purpose of this study is to assess the current published evidence regarding XZD as treatment for primary dysmenorrhoea.

**Materials and methods** The following databases will be searched from their inception until April 2017: MEDLINE (via PubMed), Allied and Complementary Medicine Database (AMED), EMBASE, The Cochrane Library, six Korean medical databases (Korean Studies Information Service System, DBPia, Oriental Medicine Advanced Searching Integrated System, Research Information Service System, Korea Med and the Korean Traditional Knowledge Portal), three Chinese medical databases (China National Knowledge Infrastructure (CNKI), Wan Fang Database and Chinese Scientific Journals Database (VIP)) and one Japanese medical database (CiNii). Randomised clinical trials (RCTs) that will be included in this systematic review comprise those that used XZD or modified XZD. The control groups in the RCTs include no treatment, placebo, conventional medication or other treatments. Trials testing XZD as an adjunct to other treatments and studies where the control group received the same treatment as the intervention group will be also included. Data extraction and risk of bias assessments will be performed by two independent reviewers. The risk of bias will be assessed with the Cochrane risk of bias tool. All statistical analyses will be conducted using Review Manager software (RevMan V.5.3.0).

**Ethics and dissemination** This systematic review will be published in a peer-reviewed journal. The review will also be disseminated electronically and in print. The review will benefit patients and practitioners in the fields of traditional and conventional medicine.

**PROSPERO registration number** CRD42016050447.

## Strengths and limitations of the study

► Our review will provide useful and novel information for patients, policymakers and practitioners.
► To avoid language bias, the Chinese, Korean and Japanese databases will be searched.
► Our systematic review will describe a comprehensive and objective assessment of the safety and effectiveness of Hyeolbuchukeo-tang/Xuefu Zhuyu decoction as treatment for patients with primary dysmenorrhoea.
► We will assess the methodological and reporting quality of included studies with the consolidated standards of reporting trials (CONSORT) extension for herbal medicine.
► One major limitation of our study protocol is that many of the included trials may have poor methodological quality or include insufficient explanation. This limitation compromises the accurate assessment of the quality of these clinical trials and their effect size. In addition, it means that insufficient information is available for future clinical trial protocol development.

## INTRODUCTION

Primary dysmenorrhoea is a common complaint that refers to painful menstrual cramps in the lower abdominal region during menstruation in the absence of an identifiable pathological condition among menstruating women.[1] Due to the different definitions of the condition, and the lack of standard methods for assessing the severity of dysmenorrhoea, prevalence estimates vary between 45% and 95% of menstruating women.[2] Dysmenorrheic pain has been reported to be the primary cause of recurrent short-term school or work absenteeism among young women of childbearing age.[3] Furthermore, dysmenorrheic pain has an immediate negative impact on quality of life, for up to a few days every month. Women with primary dysmenorrhoea have a significantly reduced quality of life, poorer mood and poorer sleep quality during menstruation compared with women who do not report dysmenorrhoea.[3]

Non-steroidal anti-inflammatory drugs (NSAIDs) are considered the primary treatment for primary dysmenorrhoea, but the quality of the evidence is low mainly due to poor reporting of study methods. In addition, NSAIDS commonly cause adverse effects, including indigestion, headaches and drowsiness.[4] Therefore, many women also seek alternative therapies to manage their menstrual discomfort including heating pads for cramps, transcutaneous electric nerve stimulation, Chinese herbal medicine (CHM) and acupuncture.[3 5–7] A Cochrane review suggested that CHM was promising for managing primary dysmenorrhoea, although the quality of the included studies was poor.[5] However, the review included all types of CHM and is outdated, requiring another study that focuses on a specific type of CHM.

In traditional Chinese medicine or Korean medicine, the main factor causing menstrual abdominal pain is blood stagnation.[8] Xuefu Zhuyu decoction (XZD) or Hyeolbuchukeo-tang was the most frequent formula used in the blood stasis researches in Korea.[9] Several systematic reviews regarding other CHM such as Danggui Shaoyao San,[10] Shaofu Zhuyu decoction[11] or Gyejibongneyong-hwan[12] have already been published or planned. However, no systematic review regarding XZD in primary dysmenorrhoea has been planned or published yet. Therefore, in this review, we will investigate current evidence related to the effectiveness of XZD or Hyeolbuchukeo-tang, a traditional herbal formula, as a treatment for primary dysmenorrhoea.

## MATERIALS AND METHODS

### Study registration

The protocol for this systematic review has been registered on PROSPERO 2016 under the number CRD42016050447.

### Data sources

The following databases will be searched from inception to April 2017: Medline (via PubMed), EMBASE (via OVID), the Cochrane Central Register of Controlled Trials (CENTRAL), Allied and Complementary Medicine Database (AMED) and Cumulative Index to Nursing and Allied Health Literature (CINAHL). We will also search six Korean medical databases (Oriental Medicine Advanced Searching Integrated System (OASIS), Korean Traditional Knowledge Portal (KTKP), Korean Studies Information Service System (KISS), Research Information Service System (RISS), KoreaMed and DBPia), three Chinese databases (China National Knowledge Infrastructure Database (CNKI), Wanfang and Chinese Scientific Journals Database (VIP)) and one Japanese medical database (CiNii). We will also search conference proceedings of relevant journals and conduct hand searching. Clinical trial registries will also be searched. The search term will be composed of the disease term part (eg, dysmenorrhoea, menstrual pain, painful menstruation, period pain, painful period, cramps, menstrual disorder, pelvic

pain) and the intervention term part (eg, Xuefu Zhuyu granule/decoction/formula/tang/capsule/pill/tablet).

The search strategies that will be applied to the Medline database and CNKI are presented in the online Supplementary material. Similar search strategies will be applied to the other databases. Study selection will be documented and summarised in a PRISMA-compliant flow chart (http://www.prisma-statement.org) (figure 1).[13]

### Types of study

All prospective randomised controlled trials will be included. However, some Chinese articles do not describe the randomisation method in detail but use only the word randomisation (随机). We will include such articles, but we will also assess the risk of bias as high if detailed randomisation processes are not described. Some articles used inappropriate randomisation processes, such as the tossing of a coin; we will exclude such articles. A cross-over design clinical trial will be also included, but only the first phase data will be presented in the effect size tables and used in the meta-analysis. A pragmatic clinical trial will also be included, based on the agreement of two reviewers (JL and JJ).

### Type of participants

Patients with primary dysmenorrhoea will be considered in the systematic review. Dysmenorrhoea secondary to other pathologies, such as uterine myoma, endometriosis or infection, will not be included in this review.

### Type of interventions

Randomised studies of the XZD formula, either as the sole treatment or as an adjunct to other treatments which were applied in both groups (intervention and control groups) in the same manner, will be included. Trials comparing XZD formula with any type of control intervention will also be included. Control group intervention could be placebo XZD, no treatment, conventional medication or other treatments. XZD is composed of 11 herbs. We will also include modified XZD, which contains less than 50% of modified herbs; if the proportion of modified herbs is more than 50%, inclusion of such compounds will be determined based on the agreement of two researchers. No language restrictions will be imposed. Hard copies of all articles will be obtained and read in full text.

### Data extraction

Two authors (JJ and JL) will perform the data extraction and quality assessment using a predefined data extraction form. The form includes information pertaining to first author, study design, language of publication, country where the trial was conducted, clinical setting, diagnostic criteria, disease duration, number of participants allocated to each group, dropout number, treatment duration, dosage of XZD, pattern identification of the participants' comparison groups, outcome, outcome results, follow-up periods, adverse events associated with XZD and composition of XZD. When studies report outcomes at more than one time

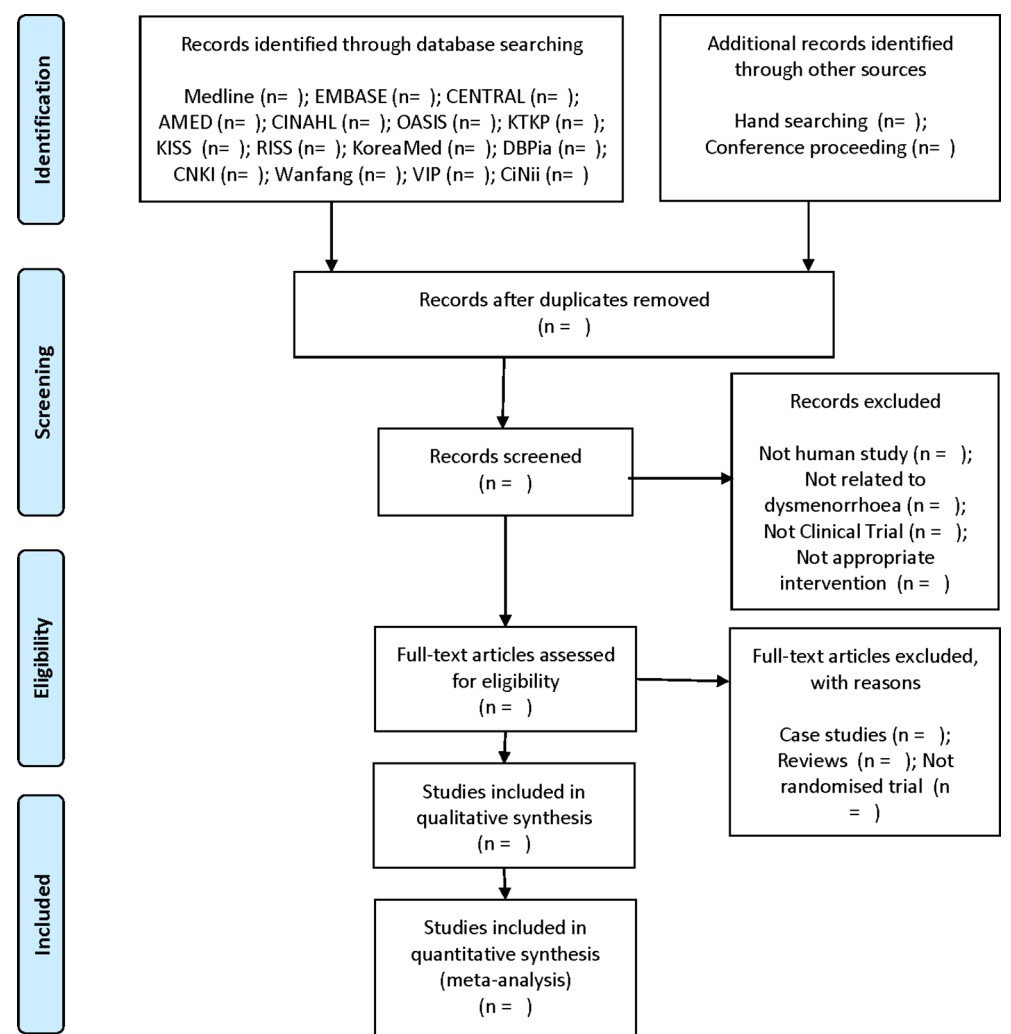

**Figure 1** PRISMA flow diagram. AMED, Allied and Complementary Medicine Database; CENTRAL, Cochrane Central Register of Controlled Trials; CINAHL, Cumulative Index to Nursing and Allied Health Literature; CNKI, China National Knowledge Infrastructure; KISS, Korean Studies Information Service System, KTKP, Korean Traditional Knowledge Portal; OASIS, Oriental Medicine Advanced Searching Integrated System; RCTs, randomised clinical trials; RISS, Research Information Service System.

point, a similar measurement point in other studies will be obtained for analysis. Any disagreement among the authors will be resolved by discussion among all of the authors. When the data are insufficient or ambiguous, JL will contact the corresponding authors by email or telephone to request additional information or clarification.

### Assessment of risk of bias in included studies

We will assess risk of bias in included studies according to risk of bias assessment tool in Cochrane Handbook.[14]

Risk of bias in included studies will be classified into three categories (low, unclear and high) by two independent reviewers. We will assess selective reporting, incomplete outcome data, blinding of the participants and personnel, blinding of the outcome assessments, allocation concealment, random sequence generation and other sources of bias.[14] Disagreements between the two reviewers will be resolved by final decision of the arbiter (KP).

### Outcome measures

#### Primary outcomes

► Change in symptoms as indicated on a 100 mm visual analogue scale
► Response rate: an overall reduction in symptoms (menstruation-related symptoms including dysmenorrhoea)

As most Chinese trials report outcomes based on a categorical assessment (eg, 'markedly improved', 'improved', 'slightly better' or 'no effect'), we will evaluate the response rates by three different methods because variation in effectiveness evaluation creates variation in results: (1) We will classify the 'no effect' category as non-responder and other categories as responder. For example, if a treatment group of 100 women are measured for intensity of pain using markedly improved (n=30), moderately improved (n=40), slightly better (n=20) or no reduction (n=10), then the number of women who report any reduction (n=90) will be considered as the responder

group and included in the meta-analysis as having experienced a reduction in pain (n=90/100). (2) Improvement in symptoms by >50% will be classified as responder; an improvement of <50% will be classified as non-responder. If the criteria for categorical assessment are not described or are unmatched, that article will not be included in the analysis. (3) We will classify the categories 'no change' and 'worsening of symptoms' as non-responder; we will classify the category 'shows improvement' as responder.

### Secondary outcomes
► Quality of life as measured using validated questionnaires
► Adverse events

### Data synthesis and analysis
In order to help researchers, the effect size of every outcome in each clinical trial will be presented for future clinical trial protocol development. Statistical analyses will be performed with the Review Manager program (V.5.3 Copenhagen: The Nordic Cochrane Centre, The Cochrane Collaboration, 2014). Trials will be combined according to the type of intervention and type of outcome measure and/or control. Data will be pooled and expressed as mean differences or standardised mean difference for continuous outcomes and risk ratio for dichotomous outcomes with 95% CIs using fixed or random-effects models.

### Dealing with missing data
As much as possible, we will analyse the data using an intention-to-treat basis, and we will attempt to obtain missing data from the original investigators. If these attempts are not successful, we will not impute data for missing data; we will analyse only the available data.

### Assessment and investigation of heterogeneity
Heterogeneity among studies will be assessed using $\chi^2$ test with a significance level of p<0.1 and $I^2$ statistic.[15] The $I^2$ statistic indicates the proportion of variability among trials that is not explained by chance alone, and we consider an $I^2$ value >50% to indicate a substantial heterogeneity.[15 16] If substantial heterogeneity is detected, we will explore sources of heterogeneity by performing subgroup analysis. If some factors (eg, lack of included trials, large methodological and/or clinical difference among trials) are found, we will not conduct subgroup analysis or data synthesis, but report a narrative description of the included studies. Subgroup analyses will be attempted according to type of control (eg, kind of medicine), taking into consideration the characteristics of the included studies.

### Subgroup analysis
If a sufficient number of subgroup studies exist, subgroup analysis will be conducted to identify heterogeneity between subgroups. Subgroup analysis criteria are as follows: (1) duration or dosage level of herbal medicine

treatment; (2) type of control intervention: placebo XZD, no treatment or western medication; (3) duration or severity of primary dysmenorrhea; (4) pattern identification according to TCM theory and (5) physical form of XZD, that is, decoctions, granules or pills.

### Sensitivity analysis
Methodological and reporting quality of included studies will be assessed by the consolidated standards of reporting trials (CONSORT) extension for herbal interventions.[17] To identify the robustness of the meta-analysis result, sensitivity analysis will be conducted after excluding low-quality trials. We will compare original and sensitivity meta-analysis results.

### Assessment of reporting biases
When there are more than 10 trials in the analysis, reporting biases such as publication bias will be assessed by funnel plots. If asymmetry is suggested by a visual inspection, we will perform exploratory analyses using Egger's method.[15]

## DISCUSSION
### Ethics and dissemination
The purpose of our review is to assess the effectiveness and safety of XZD in women with primary dysmenorrhoea. Several systematic reviews of CHM have already been published.[5 10 11] Even though XZD is frequently used in primary dysmenorrhoea,[18] no systematic reviews on the effects of XZD formula on primary dysmenorrhoea have been published. This systematic review will provide a summary of the current evidence related to the effectiveness of XZD formula for the treatment of primary dysmenorrhoea. In particular, we will identify subtypes that are particularly useful for specific subgroups according to Traditional Chinese Medicine (TCM) theory or TCM pattern identification. We will also identify a range of dosages and modifications used to improve effectiveness in full review of this protocol. We know that most of the systematic reviews in the field of traditional medicine have drawn the conclusion that 'there is some supporting evidence for the use of herbal medication but the methodological and reporting quality are both poor'. We believe that the purpose of a systematic review is not simply the mathematical synthesis of existing clinical trial results but also to offer detailed information relevant to clinical trial protocol development and clinical practice. Accordingly, we will show the effect size for all clinical trials to help researchers and physicians. Detailed information on the clinical trial regimens of XZD in primary dysmenorrhea will also provide an insight to researchers who are planning XZD clinical trials on this subject. We also anticipate finding predicting factors of treatment response by subgroup analysis. This evidence will also be useful to medical practitioners and patients in the field of women healthcare.

This systematic review does not need ethical approval because only published data will be included in our

review. This systematic review will be published in a peer-reviewed journal. The review will also be disseminated electronically and in print. The results will be presented in international academic conferences. The review will benefit patients and practitioners in the fields of traditional and conventional medicine.

**Contributors** The study was conceptualized by JJ. The protocol was drafted by JJ and JL. The search strategy was developed by JL and JJ. JML and KSP revised the manuscript. JL submitted the manuscript for publication. All authors have read and approved the final manuscript.

**Funding** This study is supported by the Traditional Korean Medicine R&D program that is funded by the Ministry of Health & Welfare through the Korea Health Industry Development Institute (KHIDI, grant HB16C0018).

**Competing interests** None declared.

**Provenance and peer review** Not commissioned; externally peer reviewed.

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
