## [Reviewer comments · BMJ Open]

ARTICLE DETAILS

TITLE (PROVISIONAL)	Herbal medicine (Hyeolbuchukeo-tang or Xuefu Zhuyu decoction) for treating primary dysmenorrhoea: protocol for a systematic review of randomised controlled trials
AUTHORS	Leem, Jungtae; Jo, Junyoung; Lee, Jin Moo; PARK, KYOUNG SUN

VERSION 1 - REVIEW

REVIEWER	Deborah Kennedy, Research Associate Canadian College of Naturopathic Medicine, Canada
REVIEW RETURNED	31-Jan-2017

GENERAL COMMENTS	page 5 In 17 the authors state "Studies where the control group received the same treatment as the intervention group will also be included." This is not clear. If both group receive the same treatment then does this not defeat the purpose? Perhaps this is just about more clearly wording the statement. page 8 In 8 The author discuss subgroup analysis by TCM Pattern however, subgroup analysis is only planned, in the protocol, if there is heterogeneity. I believe that it would be beneficial to include the subgroup analysis on TCM pattern type regardless of the presence of heterogeneity. re: The Medline search strategy - I noticed that in searching for the TCM patent name - the herbal name can be written as Xuefu Zhuyu or Xue fu Zhu yu. It maybe be a good strategy to include the various spellings in the search to ensure that all relevant studies are identified.
---

REVIEWER	Andrew Flower University of Southampton, UK
REVIEW RETURNED	21-Feb-2017

GENERAL COMMENTS	This is a well written fairly standard systematic review protocol. I think the paper could be improved by more discussion around the limitation of this approach. For example: 1. describing difficulties about deciding whether a trial has been properly randomised (reporting is notoriously poor).2. how will you accommodate papers where the basic formula has been modified?3. In all probability you will exclude a large number of papers and end up with a review providing preliminary evidence that requires more rigorous proof...are there alternative ways of exploring the data eg Evidence Synthesis?
---

	A couple of small changes would help the existing text: P 6 118 You state 'Studies where the control group received the same treatment as the intervention group will also be included. ?' This is confusing...do you mean control groups who receive herbal medicine as a control? p8 sub group analysis: what about including dosage, means of administration (decoctions/granules/pills), and TCM subgroups (eg Blood stasis + Cold vs Blood stasis + Liver Qi stagnation) as possible sub groups for analysis?
--	--

REVIEWER	Ju Ah Lee Korea Institute of Oriental Medicine South Korea
REVIEW RETURNED	23-Feb-2017

GENERAL COMMENTS	This is a well written, fairly typical, Cochrane style systematic review protocol. And there are some advices as follow:  1) The literature review has to be done after March 2016 instead of October 2016. 2) In method part, 'type of study' has to be considered in detail. Are you going to include any RCTs? I wonder how authors deal with cross over design or pramitic trials. 3) In type of interventions, will be allowed any modified XZD formula? 4) I suggest extracting an item of the criteria effectiveness evaluation, because different effectiveness evaluation could make different results. 5) My consideration with this kind of protocol is that it will almost certainly produce the same story as all the multitude of previous systematic review of herbal treatments, namely, 'there is some supporting evidence the use of HM but the methodological quality and reporting quality are poor. Further studies are needed.' Authors have to try extract valuable informations for HM use. Thank you and good luck!
---

VERSION 1 – AUTHOR RESPONSE

Reviewer: 1

#1

page 5 In 17 the authors state "Studies where the control group received the same treatment as the i ntervention group will also be included." This is not clear. If both group receive the same treatment then does this not defeat the purpose? Perhaps this is just about more clearly wording the statement.
ANSWER> Thank you for your helpful comment. We have presented this information more clearly in the revised version of the manuscript.

#2

page 8 In 8 The author discuss subgroup analysis by TCM Pattern however, subgroup analysis is only

planned, in the protocol, if there is heterogeneity. I believe that it would be beneficial to include the subgroup analysis on TCM pattern type regardless of the presence of heterogeneity.

ANSWER> Thank you for your helpful comment. We have inserted the following in the revised version of the manuscript: If a sufficient number of subgroups is present, we will conduct subgroup analysis regardless of the presence of heterogeneity. We have added subgroup analysis according to pattern identification.

#3

re: The Medline search strategy - I noticed that in searching for the TCM patent name - the herbal name can be written as Xuefu Zhuyu or Xue fu Zhu yu. It maybe be a good strategy to include the various spellings in the search to ensure that all relevant studies are identified.

ANSWER> Thank you for your helpful comment. We have accordingly revised the search strategy in the supplementary file.

Reviewer: 2

This is a well written fairly standard systematic review protocol. I think the paper could be improved by more discussion around the limitation of this approach. For example:

#1.

describing difficulties about deciding whether a trial has been properly randomised (reporting is notoriously poor).

ANSWER> Thank you for your valuable advice aimed at improving our manuscript. We have revised this issue in more detail in the section on types of studies. We will include such studies if these include randomization (随机), but we will grade the risk of bias assessment as high if no detailed randomization process is described. However, if an incorrect randomization method was used and is described clearly in Methods, that study would be excluded.

#2.

how will you accommodate papers where the basic formula has been modified?

ANSWER> Thank you for your valuable advice aimed at improving our manuscript. We will now include modified XZD. We have added detailed criteria for modified XZD in the section on types of intervention.

#3.

In all probability you will exclude a large number of papers and end up with a review providing preliminary evidence that requires more rigorous proof...are there alternative ways of exploring the data eg Evidence Synthesis?

ANSWER> I agree with you on this issue. I think that the purpose of a systematic review is not simply mathematical synthesis of the clinical trial results. It should offer appropriate information for clinical trial protocol development and clinical practice. So, we will show the effect size of every outcome and detailed regimen of intervention in the Results and Discussion, which will helpful for both researchers and physicians. I consider, in addition, that it will overcome a monotonous conclusion of systematic reviews. I have added information on this issue in the section on data synthesis/analysis and in the Discussion.

A couple of small changes would help the existing text:

#4 P 6 118 You state 'Studies where the control group received the same treatment as the intervention group will also be included. ?'

This is confusing...do you mean control groups who receive herbal medicine as a control?

ANSWER> Thank you for your helpful comment. We have amended this in the revised version of the manuscript.

#5

p8 sub group analysis:

what about including dosage, means of administration (decoctions/granules/pills), and TCM subgroups (eg Blood stasis + Cold vs Blood stasis + Liver Qi stagnation) as possible sub groups for analysis?

ANSWER> Thank you for your valuable advice aimed at improving our manuscript. I have accordingly added a further subgroup in the subgroup analysis section.

Reviewer: 3

#1 The literature review has to be done after March 2016 instead of October 2016.

ANSWER> Thank you for your helpful comment. In the revised version of the manuscript; April, 2017 is now given as the finish date in both the Abstract and main manuscript

2) In method part, 'type of study' has to be considered in detail. Are you going to include any RCTs? I wonder how authors deal with cross over design or pramitic trials.

ANSWER> Thank you for your valuable advice aimed at improving our manuscript. We will include crossover studies and pragmatic clinical trials. Detailed criteria of such trials are described in the section on types of study.

3) In type of interventions, will be allowed any modified XZD formula?

ANSWER> Thank you for your valuable advice aimed at improving our manuscript. We will include modified XZD. We have now added detailed criteria for modified XZD in the section on types of intervention.

4) I suggest extracting an item of the criteria effectiveness evaluation, because different effectiveness evaluation could make different results.

ANSWER> We totally agree with your comment. Accordingly, we have suggested three criteria for different effectiveness evaluations in the section on primary outcomes.

5) My consideration with this kind of protocol is that it will almost certainly produce the same story as all the multitude of previous systematic review of herbal treatments, namely, 'there is some supporting evidence the use of HM but the methodological quality and reporting quality are poor. Further studies are needed.' Authors have to try extract valuable informations for HM use.

ANSWER> I agree with you on this issue. I think that the purpose of a systematic review is not simply mathematical synthesis of the clinical trial results. It should offer appropriate information for clinical trial protocol development and clinical practice. Therefore, we will show the effect size of every outcome and detailed regimen of intervention in the Results and Discussion, which will helpful for both researchers and physicians. I consider, in addition, that it will overcome a monotonous conclusion of systematic reviews. I have added information on this issue in the section on data synthesis/analysis and in the Discussion.

VERSION 2 – REVIEW

REVIEWER	Deborah Kennedy, ND PhD Research Associate The Canadian College of Naturopathic Medicine, Toronto, Canada
REVIEW RETURNED	10-Apr-2017

GENERAL COMMENTS	No additional comments to make on the revisions.
--

REVIEWER	Ju Ah Lee Korea Institute of Oriental Medicine, South Korea
REVIEW RETURNED	03-Apr-2017

GENERAL COMMENTS	That will be a good review for traditional medicine.
--